# Efficiency in the Last Mile of Autonomous Ground Vehicles with Lockers: From Conventional to Renewable Energy Transport

Olga Levkovych and Adriana Saraceni *

School of Business and Economics, Maastricht University, 6211 LK Maastricht, The Netherlands
* Correspondence: a.saraceni@maastrichtuniversity.nl

**Abstract:** This research aims to compare autonomous ground vehicles with conventional and electric vans on the basis of associated vehicle costs and benefits related to their use, taking into account economic feasibility. Cost per vehicle kilometre is derived using the total cost of ownership method adjusted with the inclusion of labour costs and the impact of solar panel application on fuel efficiency while travel time-related and capacity occupations and reliability benefits serve as a basis for the total possible number of parcels delivered. The results show that, under the current structural and infrastructural conditions of urban delivery, the experimental model can be potentially successful in terms of cost per kilometre (0.133/km) but not as effective in terms of the total possible number of parcels delivered. This study defines autonomous ground vehicles with lockers as an innovative last mile solution and contributes to the academic literature by investigating the concept's efficiency competitiveness.

**Keywords:** total cost of ownership; autonomous ground vehicles; last-mile delivery; electric cars; solar panels

## 1. Introduction

The collocation "last mile" indicates the final leg of technology that connects an end-user to the rest of the internet network in telecommunications [1]. In the logistics sector, the term refers to direct end-customer interaction [2], while last mile logistics (LML) is experiencing enhanced attention from the scientific community [3]. This interest is related to trends in purchasing behaviour and demographics, as well as increasing challenges that practitioners face while trying to keep up with the basic LML principle: minimizing the freight movement required to satisfy demand while minimizing costs and negative impact [4].

LML is affected by the growth of e-commerce, given that massive internet penetration has allowed retail companies to discover new business models and engage with the buyer as directly as possible, regardless of his geographical location. The COVID-19 pandemic induced an additional shift towards online sales [5], estimating e-retail sales to reach 21.8% of all retail sales in 2024, namely 6.5 trillion US dollars in the absolute state [6]. For logistics service providers (LSPs), an increase in direct-to-consumer deliveries means an increase in freight movements, resulting in higher costs and the danger of customer dissatisfaction associated with insolvency to meet growing delivery standards [7,8]. Nowadays, 55% of the population is living in the cities [9], and the ratio is expected to rise to 65–70% by 2050 [10,11]. Rising density in urban regions entails logistical challenges associated with the delivery volume increase, the location of consolidation centres, and vehicle routing in congested areas [12]. On top of that, the ageing workforce in industrialized countries contributes to an increasing labour shortage, especially in physically demanding and low-payment environments like parcel delivery [13]. The increase in global parcel traffic, especially in cities with congestion, air and noise pollution, creates additional

burdens considering climate change and sustainability, as transport is the most problematic emitting sector in Europe [14]. Responsive governmental legislation aims to improve "health and the global climate" [15] along with slowly increasing customers' sustainable awareness [16], forcing LSPs to consider new delivery solutions, taking into account their impact on the planet and people while maintaining economic expediency and competition. One of the possible new solutions in the logistics industry became the emergence of advanced autonomous systems supported by recent developments in the field of electrification, artificial intelligence and technology. While three categories of autonomous last-mile delivery robots have been described in the literature most regularly, Wheeled Sidewalk Pods (Droids), Drones, and Autonomous Ground Vehicles (AGVs) with lockers [17,18], the last provide a multitude of advantages for large-scale urban parcel delivery capable of simultaneously addressing customer satisfaction, sustainability, and cost-efficiency challenges [19].

Existing research on the topic offers either narrow targeted improvements of the last-mile delivery (LMD) process, such as a routing optimization [20] or focus on a unidimensional analysis of alternatives within a thematic class, for instance, transport mode [21] or delivery system's externalities [22,23]. In relation to autonomous ground vehicles (AGVs), studies are mostly devoted to adoption acceptance [24], and environmental [25] or efficiency assessments [26], neglecting supplementary beneficial or counterbalancing factors that could provide a more holistic picture and fuller understanding of the context. With the aim to compare autonomous ground vehicles with conventional and electric vans, on the basis of the associated vehicle costs and benefits related to their use, this research seeks to answer "What are the cost–benefit segments of last-mile delivery process using Autonomous Ground Vehicles (AGVs) in urban areas?". A monetary comparison will be conducted, between an AGV and a conventional and an electric van by analysing the main cost and benefit vehicle-related segments. The outcomes can assist 3rd party LSPs as a structural example for multifaceted comparative analysis among a separate class of future robotic solutions, namely an AGV and other conventional transportation alternatives.

This research is structured as follows. Firstly, the concept of LML will be introduced, followed by a review of relevant academic perspectives and classification approaches. Secondly, in the methodology section, the main cost and benefit segments used in the analysis will be outlined. In the third section, the vehicle comparative analysis will be conducted. A sensitivity analysis of cost and benefit segments will also be performed to enhance the robustness of the study and detect additional enriching conditions for AGV deployment. In the fourth part, results are presented and discussed in the context of AGVs' potential implementation for urban delivery, while highlighting future research opportunities and practical recommendations.

## 2. Literature Review

### 2.1. Efficiency in Last-Mile Logistics

Mounting research focuses on LMD efficiency, either in terms of direct [27] or indirect costs and externalities [23,28]. Such growing interest is partly related to the spread of home delivery, which hinders the exploitation of economies of scale or marginal benefit models. LMD is considered one of the least efficient and most expensive sections of the entire supply chain [29], accounting for up to 75% of total logistics costs [30]. Nonetheless, companies typically charge less than the order's cost to fulfil it, while customers wish to pay even less than the current cost [29]. Challenges are ample, like the "customer not at home" issue, poor predictability of delivery time, inability to choose a delivery window opposite to excess mileage due to limited availability, low customer density, returns and sustainability concerns [31–34]. Most of the literature on LMD efficiency thus examines its transport component and can be conditionally divided into two categories: optimization of traditional delivery modes and innovative solutions.

Efficiency is predominant in this study, especially costs and supplement cost-related benefits. The profitability of any chosen solution or significant savings should be compared to a

baseline situation. Among the papers reviewed, the vehicle routing problem (VRP) is central due to its longstanding existence and broad application scope. Researchers have examined cost minimization of route re-planning [35], path optimization related to time windows [36], or departure time [37], and customer density [38,39]. Others explore optimization possibilities via order consolidation [19,40], or the most efficient parcel location inside a truck [41]. Such approaches are widely used by LSPs like FedEx, UPS, and DHL for higher-level decision-making and planning support for everyday tours and vehicle loading.

### 2.2. Traditional Delivery Mode Optimization and Innovations

Innovative solutions aim to overcome traditional effectiveness hurdles in a way that may improve customer satisfaction, reduce costs or even discover new business models. The VRP is highly used when analysing options like drone delivery [42] or cost minimization capabilities depending on customer-to-locker assignment [43]. Some papers examine new options [44,45] or new against regular delivery modes [46,47]. Nonetheless, many stumbling blocks remain, for instance, consumer and government reluctance to implement innovations, lack of the necessary technological background, and the issue of the uncertain paybacks from the optimization obtained by innovation [48–50]. Boysen et al. (2020) [51] define conventional vans, cargo bikes and self-service stations as status-quo contemporary methods, while drones, autonomous delivery robots (bots or droids), crowdsourcing and reception boxes as near-future concepts. For the further future, ideas on flying warehouses, mobile parcel lockers and autonomous vehicles are presented. Schröder et al. (2018) [52] offer the technology maturity overview with four time horizons, the first representing the current LMD transformation, driven by vehicle electrification and unattended delivery. The second horizon brings semiautonomous delivery vehicles, which in approximately ten years will be supplemented by fully autonomous vehicles and drones during the third horizon. The fourth horizon goes beyond 2030 and only provides optimistic hopes for technologies addressing "the last ten yards" of delivery. Figure 1 illustrates the variety of innovative delivery solutions in accordance with the four time horizons.

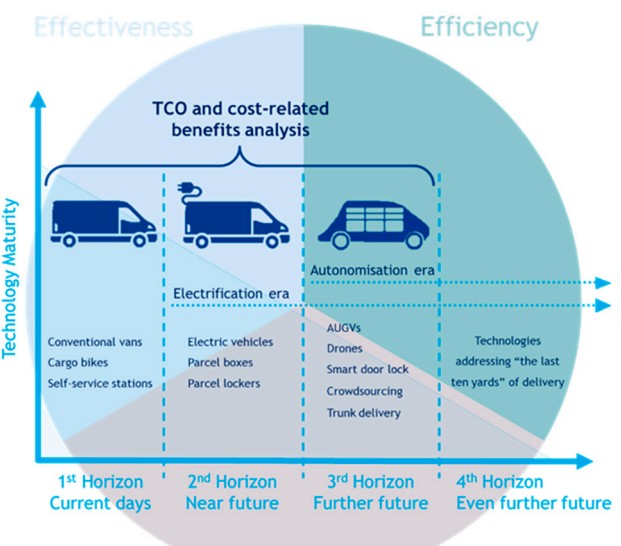

**Figure 1.** Conceptual framework based on Boysen et al. (2020) [51] and Schröder et al. (2018) [52].

### 2.3. Total Cost of Ownership Analysis and Cost-Related Benefits

Fleet electrification is the main short-term challenge for many leading LSPs based on the current technology maturity. As the penetration rate of any technology highly depends on associated costs, the increasing supply of affordable electric energy [53,54] and decreasing battery costs, the most expensive component of electric vehicles (EV) [55,56], became

the main drivers of fuel transition. While many articles highlight the environmental advantages of electric transport compared to fossil fuel emissions, this paper is primarily interested in the economic performance and cost-related benefits evaluation of certain vehicles, hence in quantitative methodologies allowing for results in monetary units [57].

Several papers advocate for the advantages of using electric vehicles [58–60] and their cost competitiveness in all mobility categories, i.e., 2-wheelers, cars, buses, and heavy and light vehicles [61]. Other studies focus on defining the conditions and beneficial factors contributing to EV cost competitiveness [62]. Among those reviewed, total cost of ownership (TCO) is a widely used method for understanding the potential benefits of a new technology [62–65]. Based on a TCO analysis, the most advantageous technology in terms of cost can be chosen, while the most problematic cost areas in need of improvement can be identified. The method summarizes all costs related to ownership of a given subject during a certain period of ownership. There are usually four types of input information needed to conduct a TCO on a transport vehicle, namely ownership period, travel data, vehicle data and cost data (for a detailed overview, see Appendix A).

Concurrently to costs, innovative technology adoption should foster optimization along with sustainability and effectiveness benefits related to fuel efficiency, travel time-related, capacity occupation and reliability improvements of costs [66]. Benefits also depend on specific vehicle characteristics and dimensions but can be further stimulated by ancillary mechanisms and methods, for instance, congestion mitigation, de-emphasized performance, vehicle right-sizing and light-weighting, eco-driving and eco-routing, and higher speed limits. Other possible benefits can be related to the factors mentioned under the cost data of TCO analysis [67,68], for instance, a 10% discount on insurance is offered if a car has a collision-avoidance system in the UK [69]. Similar discounts could be implemented for AGVs, as they eliminate human error, increase safety, and thereby reduce the chances of an accident [70]. The TCO framework for the input data and TCO assessment algorithm was inspired by the work of Siragusa et al. 2020 [61].

## 3. Methodology

The TCO method will be used for three different types of transport. As a baseline, a conventional van with an internal combustion engine representing the current state of LMD will be used; an electric van as a common solution of the near future; and an autonomous delivery robot (ADR) as a further future potential solution. Additionally, each of the options will be assessed with the use of solar panels as an auxiliary tool for cost savings. The data were collected taking into account the specifics of urban delivery, based on resources and institutions operating in the Netherlands, and therefore reflects the context of the Dutch LMD market.

This section describes the data collection mechanisms for each of the variables needed to calculate the total actual costs and kilometres driven during the entire ownership period. The algorithm proposed by Siragusa et al. (2020) [61] is taken as a TCO input framework and includes variables such as ownership period, travel data, vehicle characteristics, and associated capital and operational costs. The results of the TCO assessment will be obtained using Formula (1) proposed by Siragusa et al. (2020) [61] and adjusted to include solar panel costs and labour costs for a delivery van driver, a substantial part of the LMD cost [52] which in ADRs will be eliminated.

$$TCO_T = PP_0 - RV_T + RFC_0 - S_0$$
$$+ \sum_{t=0}^{T} \frac{FC_t + IC_t + MRC_t + BC_t + OC_t + RT_t - II_t}{(1+i)^t} \tag{1}$$

Costs will be adjusted annually according to the discount rate of the present discounted value shown in Formula (2) [71].

$$PV = A_t \times \frac{1}{(1+I)^t} \tag{2}$$

where

$PV$ = Present value;
$A_t = Amount\ of\ one-time\ cost\ at\ a\ time^t$;
$I = Real\ discount\ rate$;
$T$ = Time (expressed as number of years).

 The total possible parcels delivered with the total cost per kilometre (km) gained from TCO are two representative metrics of delivery performance. This research will examine solar panels' fuel-saving capabilities and their effect on the cost per km for each vehicle, as well as autonomy impact on the total possible number of parcels delivered due to operational time increase. Subsequently, a sensitivity analysis of cost and benefit segments will follow, allowing for a deeper analysis and robustness check.

### 3.1. TCO Calculation

 Within the TCO calculation algorithm, the following main blocks of data are to be considered: vehicle characteristics, ownership period, travel data and labour costs.

### 3.1.1. Vehicle Characteristics

1. Vehicle overview: Vehicle type largely affects the remaining variables, and therefore requires a primary outline. PostNL, the main parcel delivery service provider in the Netherlands [72], made a deal with Renault group on the purchase of light commercial electric vehicles, Master Z.E (L3H2) [73], as a part of its plan for emission-free LMD by 2030 [74]. The same model exists in a version with the diesel engine, which provides a convenient comparison basis within a research's conceptual framework. The autonomous ground vehicle (AGV) with lockers, namely Neolix, which McKinsey judges as a promising development for replacing some of the current urban delivery modes [17], was selected as the future delivery option. Neolix was deployed in China by tech and logistics giants such as Huawei, Alibaba, Meituan-Dianping, and JD [75], who collaborated in a trial with Swiss Post in Europe [76], and piloted autonomous deliveries for e-commerce startup Noon.com in the Middle East [77]. Table 1 includes the overview of the information that will later be needed to define various main segments of cost–benefit analysis. Purchasing price and curb weight are used to assess eligibility for subsidies as well as define the exact rate of various ownership costs for our analysis. Motor type data, together with vehicle fuel consumption, are used in calculations to determine the fuel efficiency impact (cost savings) provided by SolarOnTop (SOT) technology. Payload, maximum speed, and maximum range serve as input to calculate benefit or maximum daily delivery volumes per vehicle. The starting price, which assumes the inclusion of battery cost, and an overview of the vehicles can be found in Table 1, whereas the rest of the descriptive information, retrieved from Renault.nl [78], Neolix.ai [79], and the department of road transport-RDW [80], can be found in Appendix B.

Table 1. Vehicle overview.

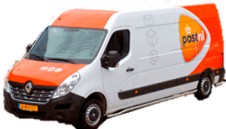 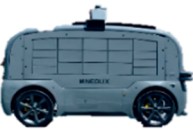

| Vehicle Name | Motor Type | Payload $m^3$ | Curb Weight (kg) | Max. Speed (km/h) | Purchase Price (EUR, excl. VAT) | Max. Range (km) | Fuel Consumption (l/100 km or kWh/100 km |
|---|---|---|---|---|---|---|---|
| **Renault Master Diesel** | 2.3 L diesel engine | 13.0 | 2066 | 148 | 31,940 | 1200 | 6.7 |
| **Renault Master Z.E.** | 5.7 kW BEV with 33 kWh battery | 13.0 | 2050 | 100 | 58,700 | 120 | 27.5 |
| **Neolix Express** | BEV with swappable 12.9 kWh battery | 2.4 | 371 | 50 | 30,000 | 100 | 12.9 |

Sources-Renault.nl [78], Neolix.ai [79], Japan-times.co.jp [81].

2.  Solar panels: Current technology maturity does not allow for commercial vehicles running entirely on solar power, yet there are few examples of solar panels providing fuel savings and extending a vehicle's battery life [82,83]. Dutch company IM Efficiency has developed the SolarOnTop (SOT) product, providing trucks with clean electricity generated by solar panels otherwise generated by the alternator [84], reducing the load on the engine and hence fuel consumption, thus preventing the costly idling hours. All SOT-related information was obtained during an informal interview with a company representative. The SOT price is determined separately for each vehicle.

3.  Depreciation rate: According to Dutch law, the depreciation rate for delivery vans is 100% after five years [85]. Due to the lack of similar information concerning the ADRs, the same depreciation rule will be applied to the AGV with lockers, considering the residual value of EUR 0 in all cases (see Table 1).

4.  Vehicle-related costs, fees and taxes: Upon purchase, a vehicle owner is automatically responsible for registration, insurance, APK (Algemene Periodieke Keuring), and motor vehicle tax (see Appendix A).

5.  Vehicle-related costs, subsidies and indirect initiatives: According to the Netherlands Enterprise Agency–RVO [86], there is a list of financial support for businesses driving electric with a minimum reliability level of 2+ where 'RVO services' authorization is assumed in order to receive these subsidies. Allowances are applied to corresponding purchase costs and can be combined except for the MIA and EIA combination. Those able to be applied to this research are presented in detail in Appendix A.

### 3.1.2. Ownership Period

1. Period of ownership: The transport industry traditionally operates on low margins and therefore tends to exploit its assets on the maximum to minimize operational costs distributed over the asset's lifespan [87]. According to Topsector Logistics (2017) [88], transport industry standards limit the recommended period of usage for a commercial van to 8 years. The same number of years is proposed by the European Environment Agency.

2. Discount rate: Some papers devoted to TCO on vehicles use a national long-term interest rate [52] while others use a long-term governmental bond rate [65,89] as a discount rate. For the Netherlands, both of these rates are negative [90,91], indicating upcoming inflation and thus working as a stimulus to invest while not adequately representing the time value of money or risk on return. Based on these considerations, the real discount rate

was set at 0,05% to eliminate the complexity of inflation consideration within the present value calculations.

### 3.1.3. Travel Data

Information on the specific annual mileage numbers for vans used by the PostNL parcel delivery subdivision is confidential. However, the approximate annual mileage value for parcel transporters like PostNL, DHL, and UPS can be retrieved from the Central Bureau of Statistics of the Netherlands (CBS) and RDW [80]. In the Netherlands, 3rd party logistics service providers are registered under the SBI code "53: Postal and courier activities" [92]. The average annual mileage for these sectors is 38,753 km [88] which matches the annual mileage (39.082 km) that can be calculated using the case study within the outlook of the 'Package market and home deliveries' city logistics segment in the MRDH region [93]. Accordingly, an annual mileage amount of 39,000 km was chosen.

### 3.1.4. Labour Costs

The average hourly salary for a parcel delivery employee in PostNL is EUR 10.79, according to the last update on 25 May 2021, on the Indeed.com (2021) [94] website, which is in line with wage information under parcel deliverer vacancy at PostNL (2021) [95]—EUR 1973 gross per month assuming 23 working days, 8 h per shift.

### *3.2. Benefits Calculation*

### 3.2.1. Capacity Occupation and Reliability

LSPs are also concerned about the maximum possible number of items delivered. In November 2020, Topsector Logistiek [96] conducted an analysis on parcel and home delivery at the MRDH area and noted 75,000 parcels delivered daily by 340 vans, which equals to 220 parcels per car. The study also found that the capacity of one van is 300 parcels and it usually has an average load factor of 73%, which corroborates the 220 parcels value per van. With van delivery, the driver takes the parcel from the cargo compartment and brings it to the recipient, while during autonomous delivery, a single cargo compartment can pose a danger to the safety of parcels, namely the risk of theft. Therefore, to the detriment of the total available cargo space, this research will assess a cargo compartment divided into separate lockers. Neolix offers three different configurations of lockers, namely the following:

Big locker dimensions: 420 × 280 × 510 mm (59 L); Medium locker dimensions: 420 × 245 × 125 mm (15 L); Small locker dimensions: 420 × 280 × 83.3 mm (9.7 L); Total locker volume: 471.5 L.

Companies engaged in B2C delivery in 90% of cases have their sorting centres dealing with small- and medium-sized parcels [97], while at the same time, 85% of e-commerce purchases in 2020 had a weight of less than 2 kg [98]. For items less than 2 kg and with a maximum size of 380 × 265 × 32 mm, PostNL and Bol.com use Letterbox Packets+, an intermediate package between a letter and a parcel [99,100]. Solely from the parcel perspective, according to the study of Louter (2019) [101] exploring the last-mile parcel delivery market in Groningen, parcel sizes of 8, 13, and 18 L were among the most frequently used, with the volume distribution of 40%, 40% and 20%, respectively. Thus, taking into account the average parcel volume of 12 L and the Neolix Express total volume of 471.5 L, the maximum capacity of 39 packages per delivery cycle was obtained.

### 3.2.2. Fuel Efficiency

Energy obtained through solar panels can substitute part of the electricity needed for all vehicle types and even affect fuel consumption for vehicles with internal combustion engines (ICEV), therefore reducing feeding costs. According to the company's representative [81], 1 m$^2$ of SOT can produce about 250 kWh per year, which in the case of electric cars increases the maximum power reserve of the vehicle at a previous rate of electricity consumption. For conventional vehicles, an additional supply of electricity, along with keeping the battery charged also lowers the load on the engine, resulting in less fuel needed

to produce Tank-to-Wheel energy [102]. The SOT simulation application developed by IM Efficiency used to calculate the total fuel savings for a vehicle with a diesel engine is confidential and restricted for external disclosure.

*3.3. Travel Time: Total Operational Time Increase*

According to the service information and vacancy description at PostNL [95,103], the working activities related to the parcels' delivery take place between 7:00 and 23:00 (two shifts/day), six days a week excluding Sundays and holidays (302 working days in 2021), while deliveries can be expected between 8:00 and 21:30 [104]. That is, the operating process is adjusted to the needs of human resources and relevant legislative limitations. Autonomy, by definition, eliminates the need for humans as process drivers, thereby allowing for a significant expansion of the available working hours, theoretically making possible 24/7 delivery. That automatically entails a number of potential improvements such as an increase in total delivery volumes, delivery time reduction during off-peak and night hours, and expended delivery windows management capabilities. On the other hand, when calculating the new delivery time prospects, it is critical to consider recipient availability as the primary limiting factor. In this study, the assumption on the expansion of delivery hours using Neolix to the 7:00–1:00 range is made, to calculate the total operational time increase benefit. Nonetheless, the suggestion is a presumption that requires scientific evaluation.

## 4. Data Analysis and Results
*4.1. TCO Analysis*

In this section, the comparison of the three vehicles (Table 1) representing the LMD transition process is conducted. Each vehicle is assessed in two versions: "standard" with a basic equipment set and "advanced" with the auxiliary tool, SOT. We aim to test the solar panel technology on efficiency improvement capabilities for vehicles used in the LMD, as it proved to be beneficial for long-distance transportation using trucks.

Taking into consideration vehicle dimensions (see Appendix B) and SOT energy generation capabilities, it is possible to calculate the annual amount of energy directly transformed into fuel savings for electric vehicles. It is assumed that all extra-generated energy is used primarily for vehicle movement, dispensing with the battery stage and neglecting other energy-consuming elements (i.e., radio), yet can be directly deducted when calculating feeding costs. To calculate annual fuel savings for a diesel engine vehicle using SOT, the following assumptions were made. The electric energy produced by SOT keeps the battery charged to perform constant switch off/on during stops without causing harm to the battery and engine. The rest substitutes part of the energy the van needs while driving, neglecting any supplementary energy-consuming elements. The stop/driving time ratio is determined as 3/5 during an 8 h shift. Results on energy generation and savings capabilities of SOT were provided for the MRDH region according to its annual solar activity (302 working days) (see Table 2).

**Table 2.** SolarOnTop savings performance.

| Vehicle Name | Area Suitable for SOT Installment (m$^2$) | Annual Energy Generated (kWh) | Annual Fuel Savings (L) |
|---|---|---|---|
| Renault Master Diesel | 6.88 | - | 998.73 |
| Renault Master Z.E. | 6.88 | 692.50 | - |
| Neolix Express | 3.25 | 346.25 | - |

The TCO cost segments for each vehicle are presented in Table 3. Some segments were grouped together for comprehensive purposes. For instance, subsidies and indirect initiatives are combined with purchasing cost under "Purchasing cost", and Registration fee, BPM tax, Road tax, and APK inspection are gathered under "Ownership (+ APK)". For a detailed calculation of purchasing and annual ownership costs, see Appendix C.

**Table 3.** Summary of the TCO cost segment.

| Cost Segments/Vehicle | Master Diesel | Master Diesel + SolarOnTop | Master Z.E. | Master Z.E. + SolarOnTop | Neolix | Neolix + SolarOnTop |
|---|---|---|---|---|---|---|
| **Purchasing Costs (EUR)** | | | | | | |
| Purchasing cost (EUR) | 31,940.00 | 37,290.00 | 58,700.00 | 64,050.00 | 30,000.00 | 352,500.00 |
| Subsidies and Indirect initiatives (EUR) | 5429.80 | 6412.86 | 14,392.00 | 14,766.50 | 5100.00 | 6064.69 |
| **Ownership (+AKP) (EUR)** | | | | | | |
| Registration fee (EUR) | 10.75 | 10.75 | 10.75 | 10.75 | 10.75 | 10.75 |
| BPM tax (EUR) | – | – | – | – | – | – |
| Road tax (EUR) (for 3 months) | 142 | 142 | 135 (from 2026) | 135 (from 2026) | 30 (from 2026) | 20 (from 2026) |
| APK inspection (EUR) (schedule pattern) | 52 (3-1-1) | 52 (3-1-1) | 45 (4-2-2-1) | 45 (4-2-2-1) | 35 (4-2-2-1) | 35 (4-2-2-1) |
| **Feeding cost (EUR/L, EUR/kWh)** | 1.52 | 1.52 | 0.11 | 0.11 | 0.11 | 0.11 |
| **Insurance cost (EUR/month)** | 128.03 | 128.03 | 69.84 | 69.84 | 64.00 | 64.00 |
| **Maintenance and repair (EUR/km)** | 0.052 | 0.052 | 0.019 | 0.019 | 0.019 | 0.019 |
| Battery cost (EUR) | - | - | - | - | - | - |
| Road toll (EUR) | - | - | - | - | - | - |
| **Labour cost (EUR/month)** | 1973.00 | 1973.00 | 1973.00 | 1973.00 | - | - |
| **TCO for 8 years (EUR)** | 280,501.96 | 266,450.62 | 255,894.20 | 260,261.67 | 41,574.82 | 45,566.12 |

Sources—RDW [105], Belastingdienst [95,106–109], RVO [110–112], Renault.nl [78], Japan-times.co.jp [81], Eurostat [113], Autodijk.nl [114], Unitedconsumers.com [115], Lebeau et al., 2019 [65].

The TCO values for each vehicle were obtained following formula 1 which sums up all the costs mentioned over eight years, with consideration of a 0.05% discounted rate. The TCO values were divided by the total number of km vehicles assumed to travel during the ownership period to receive the TCO per km. Figures 2 and 3 give an overview of separate cost elements in absolute and relative numbers shaping the TCO per km. SOT only gives net savings in the case of a diesel van; therefore, only for this vehicle, cost per km is lower with SOT.

Unsurprisingly, labour cost gains the biggest share, ranging from 67.37 to 73.85% for non-autonomous vehicles, followed by purchasing that on opposite to the previous cost segment varies substantially depending on vehicle type. The difference of the purchasing cost is better to analyse using absolute terms, as it prevents bias of relative comparison caused by the absence of Labour cost for a delivery van driver for Neolix. In this way, the purchasing cost for Renault Master Z.E. is bigger by 60% compared to the Renault Master diesel version (incl. subsidies). Concurrently, Neolix is only 6% cheaper than the diesel van, being, however, also 6.4 times smaller in size and 5.4 times smaller in maximum capacity. That proves that even with substantial subsidies and indirect initiatives available, electric vehicles are still a costlier choice than diesel ones, largely in terms of purchasing cost. However, generally higher purchasing cost is counterbalanced by all remaining operational costs, except labour, which are smaller for electric vehicles. Given the novelty of the technology and due to lack of information during future maintenance costs, those will be considered constant. Figure 4 shows vehicles' TCO structures based on the assumption that no human labour is needed for Neolix and that the vehicles adopting similar routing better represent the distribution of various operational costs comparing to capital purchasing cost.

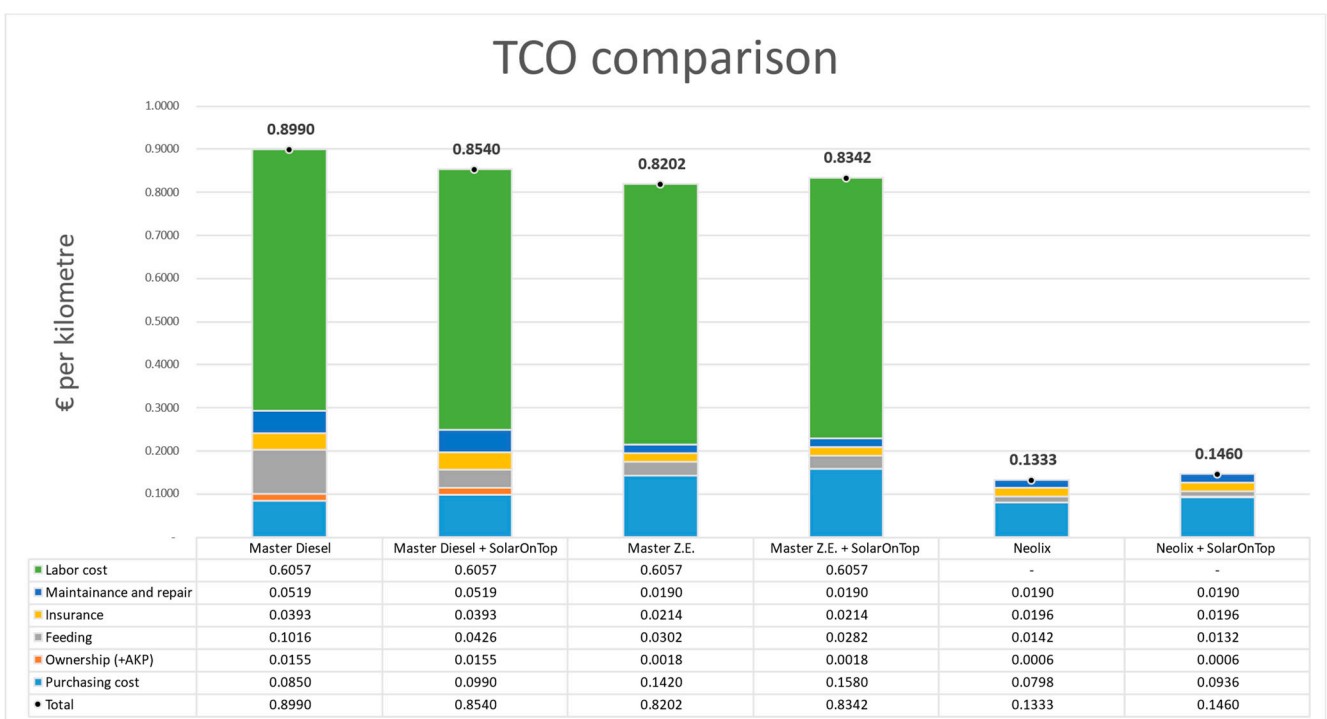

**Figure 2.** TCO per km comparison.

Maintenance and repair (M&R) absolute costs are higher for internal combustion engine vehicles, resulting in higher insurance costs, those being higher for Renault Master Diesel among all options. Interestingly, the relative cost share of M&R for an electric Neolix vehicle is higher (14.23%) than for an electric Renault Master ZE vehicle (8.84%), whereas, for insurance cost, Neolix has the highest relative cost share, obviously due to its lower purchasing cost, although having the lowest insurance cost in absolute terms. It is challenging to assess how exactly autonomy will affect M&R and insurance costs, as many factors may influence their formation, for instance, the price of autonomy-enabling equipment, security and cyber vulnerability of technology, institutional recognition of autonomous vehicles, the weight of driver behaviour, participants involved in the liability system and so on [116–118]. However, autonomy has the major advantage of eliminating labour costs, translated into EUR 23,622.5 annual savings, which can be spent to cover potentially higher absolute M&R and insurance costs. With an initial purchasing price of EUR 30,000, Neolix has a very high potential for LMD implementation solely from a cost perspective.

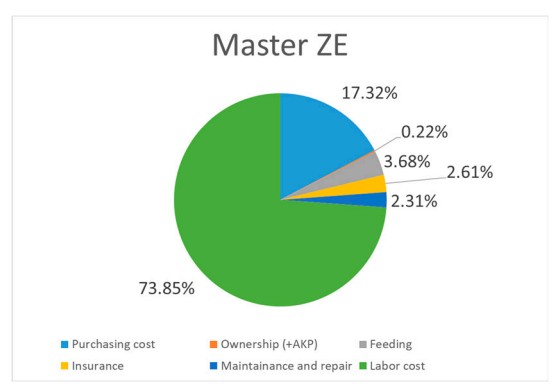
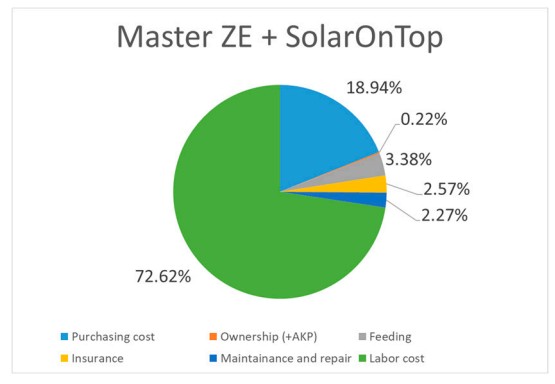

**Figure 3.** *Cont.*

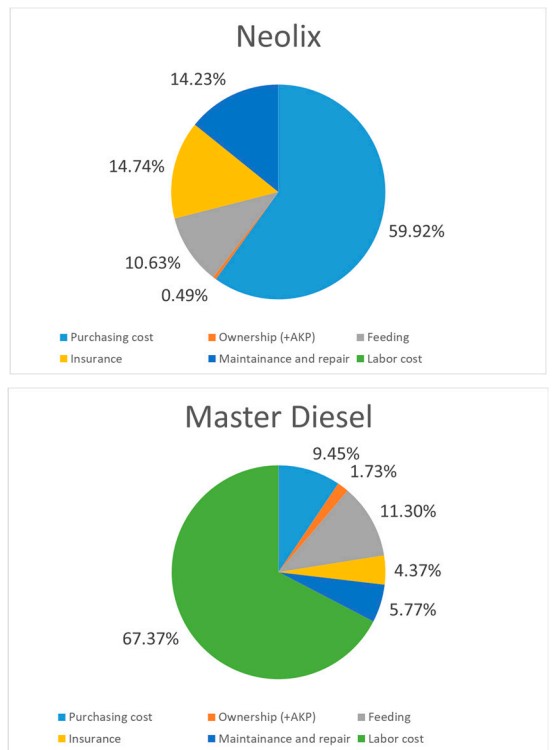

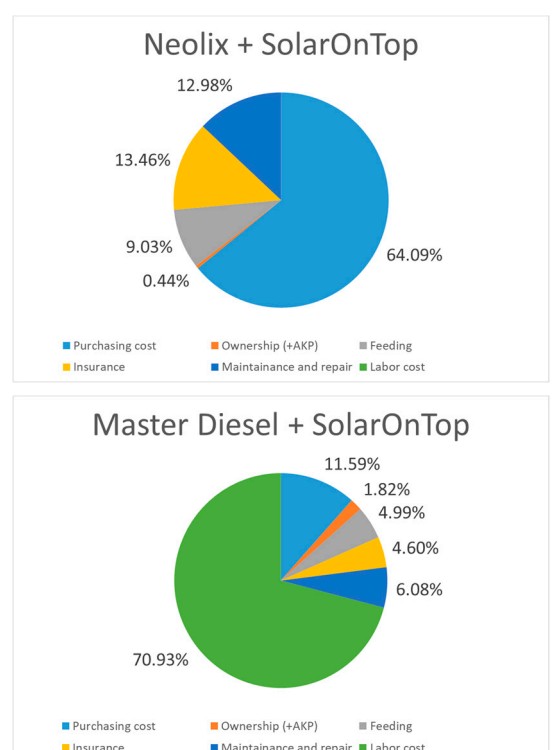

**Figure 3.** Cost distribution with labour costs.

The cost share for ownership (+APK) is much lower for electric than diesel vehicles and smaller for Neolix than for Renault Master Z.E. State support in the form of subsidies and investment allowances encourages environmentally friendlier technology, and the small weight and consumption rates of Neolix are relevant.

Regarding feeding costs, SOT did not produce significant savings for electric vehicles, not even to overcome the initial technology investment. However, for the diesel Renault Master, savings in absolute terms made up EUR 1518.07 annually, giving an approximate SOT payback period of 3.5 years. Small savings for electric vehicles can be explained by the initial higher fuel-to-wheel efficiency of EV (around 75%) compared to ICE-diesel (around 35%) [119] and the lower price of electricity.

In terms of TCO solely, the best option for LMD is the Neolix (EUR 0.1333/km), which turned out to be cheaper to use than the electric Renault Master ZE (EUR 0.8202/km) and Renault Master Diesel (EUR 0.8990/km) even without taking into account the labour costs of a van driver, which make up the bulk of all costs. SOT represents a promising application for vehicles with diesel engines, resulting in significant fuel and monetary savings.

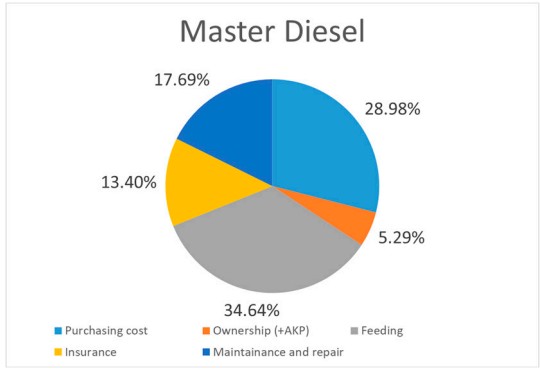

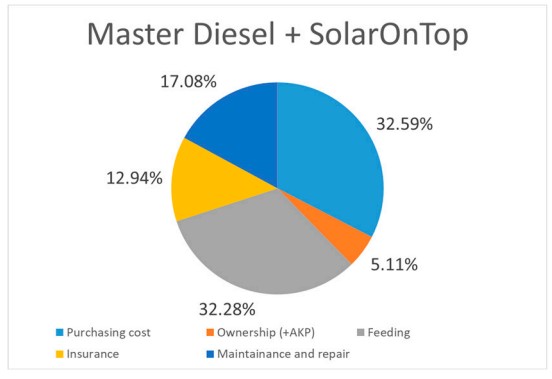

**Figure 4.** *Cont.*

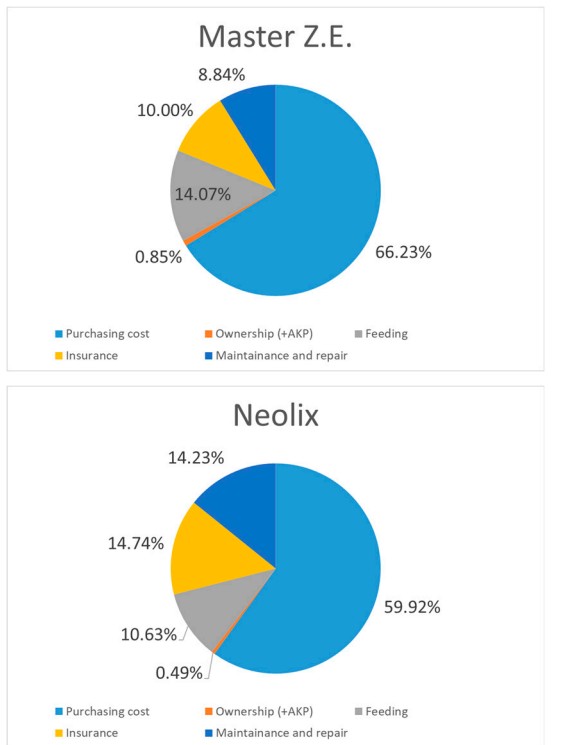
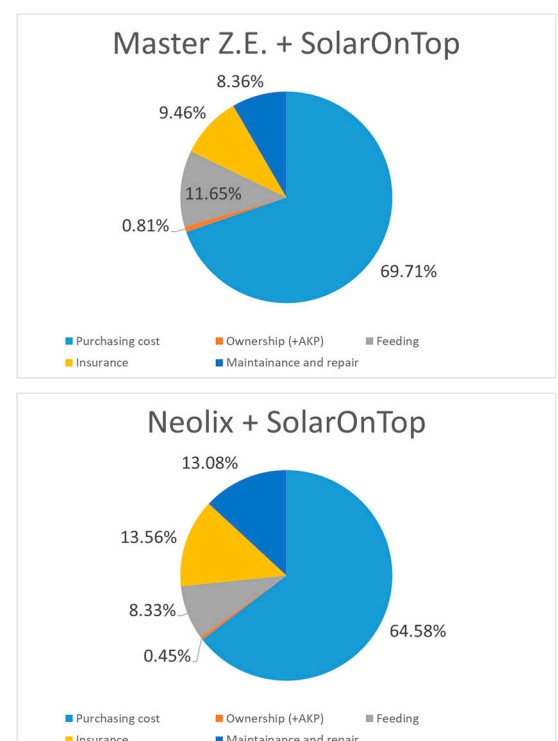

**Figure 4.** Cost distribution without labour costs.

### 4.2. Benefit Analysis

This section is devoted to a comparative analysis of different transport modes based on the total cost of ownership. However, the total number of items delivered during a specific time period is also an important performance indicator. On the opposite side of the spectrum to costs is the income constituent of the delivery profitability along with the capability to cope with the rise of parcel volumes. The vehicle's capacity occupation and the available operating time in which delivery can take place affect the total number of parcels delivered.

In this part, there will be no capacity diversification between the diesel Renault Master and its electric version. For both vans, the capacity occupation rate is based on the Parcel Market and Home Deliveries analysis of Topsector Logistics (2020) [88], where during one delivery cycle, a van is loaded with 220 units. According to PostNL, two delivery cycles are performed each day corresponding to 440 deliveries. The electric van is limited to a maximum of 1 delivery cycle per day, after which lengthy charging is needed. Therefore, two Renault Master Z.E. vans are needed versus one Renault Master with a diesel engine to cover 440 daily deliveries.

In the absence of research or other descriptive data on Neolix delivery characteristics, its operational features will be equated to those of a cargo bicycle. The given comparison is confirmed by the Minutes of the General Meeting of PostNL shareholders, in which commenting on the ban of Stint, PostNL spokesperson named diesel vans and cargo bikes as the two main alternatives for replacing Stint rides, and cargo bikes given primacy [120]. Stint resembles a halved-in-size non-autonomous version of Neolix [121], the main advantages of it being manoeuvrability, allowing it to ride in traffic-free areas, and a payload of 1.6 cubic meters [122], making it similar to cargo bikes rather than vans in terms of delivery cycle characteristics.

The mileage per package and delivery cycle time can be calculated using the main delivery characteristics of a cargo bike, i.e., capacity, delivery cycle mileage and permitted speed. The information obtained, in combination with assumptions on extended operating hours using Neolix, allows it to establish its maximum daily deliveries. Traffic lights, busy streets, cargo weight, and constant acceleration/deceleration due to frequent stops

keep the average speed of a bike courier in the urban area around 16 km/h [123,124]. DHL, the second biggest parcel delivery service in the Netherlands, uses cargo Cubicycles of a 50 km average daily mileage in Rotterdam and Den Haag [125]. Bicycle couriers usually have one shift of 3 h per day [103,126], resulting in an average delivery cycle mileage to be 50 km based on a 16 km/hour average speed. The freight volume of Cubicycle is 1 cubic meter, which at 72% utilization [101] can be equivalent to the capacity for 60 parcels with an average size of 12 L. The previous information results in a performance indicator of 0.83 km per parcel delivered. Consequently, the full delivery cycle with 39 parcels results in 32.37 km and 2 h. Following a simple calculation from Table 4, Neolix can deliver 351 parcels per day on three full charges (full charge = 100 km) that, thanks to the Neolix swappable batteries, can be carried out in less than a minute. The handling, loading and battery replacement time are not considered in this research as they are not a direct subject of interest).

**Table 4.** Maximum delivery volume calculation per vehicle.

| Renault Master | Neolix | | Neolix Delivery Performance | |
|---|---|---|---|---|
| 220 | 39 | Load capacity in units | | |
| 08:00–21:30 | 07:00–01:00 | Working hours | 16 | Vehicle speed in km/h |
| 13.5 | 18 | | 0.83 | Vehicle km per unit |
| 6 | 7 | Working days per week | 39 | Load capacity in units |
| 302 | 354 | Working days per year | 32.37 | Delivery cycle millage |
| 440 | 351 | Maximum delivery volume per day | 2 | Delivery cycle duration in hours |
| 2640 | 2457 | Maximum delivery volume per week | 9 | Number of cycles per days |
| 132,880 | 124,254 | Maximum delivery volume per year | 351 | Maximum delivery volume per day |

The tariffs that courier/postal companies charge for parcel delivery services vary, depending on the country, their local market position, the legal framework, and the degree of their network development and infrastructure spread [127]. Furthermore, cost structures differ greatly from chosen rate categories that can be based either on weight class or size or both. The domestic rates of PostNL home parcel delivery for packages 0–10 kg (max. 100 × 50 × 50 cm) will be used for income assessment (equal distribution among online and offline franking is assumed). Therefore, the income per parcel delivered is EUR 7.00 [128].

The quantitative summary of the cost and Income segments for the three vehicle types is presented in Table 5. Two Renault Master Z.E. are needed to deliver 440 daily units, as reusing one vehicle on the same day is impossible due to long recharging. Therefore, the calculated TCO per km will include an extra share of the purchase cost for a second vehicle. When considering cost-saving benefits and the income for total items delivered under the assumptions of this study, Neolix, due to its lower overall capacity and parcels per km delivered, has the lowest potential revenue per km (EUR 8.30) compared to the van delivery options of the present and near future.

For the near future, given the equal performance capabilities of both Renault Master vans, the diesel engine is a bit more profitable (EUR 11.03 revenue per km) comparing to the electric one (EUR 10.96 per km). However, due to government subsidies and lower operational costs, the TCO per km for both vehicles was almost equal, simplifying the green energy transition for LSPs and assisting smooth adaptation to governmental requirements such as 0 emission zones for urban logistics by 2025 [129].

**Table 5.** Cost–benefit summary of last-mile delivery vehicles.

| Renault Master Diesel | Renault Master Z.E. | Neolix | |
|---|---|---|---|
| 132,880 | 132,880 | 124,254 | Delivery volume per year (units) |
| 78,000 | 78,000 | 103,131 | Delivery kilometer per year (km) |
| 1.70 | 1.70 | 1.20 | Parcels per kilometer delivered (units) |
| 7.00 | 7.00 | 7.00 | Income per parcel delivered (EUR) |
| 11.93 | 11.93 | 8.43 | Income benefit per kilometer (EUR) |
| 0.90 | 0.96 | 0.13 | Cost per kilometer (EUR) |
| 11.03 | 10.96 | 8.30 | Revenue/Loss per kilometer (EUR) |

*4.3. Sensitivity Analysis*

The TCO analysis conducted in this paper for the period 2021 to 2028 contains several assumptions that imply varying degrees of uncertainty. Therefore, sensitivity analysis is performed for the vehicles without SOT to test the robustness of cost segments which either have a significant variability potential or segments whose change vector is known yet the degree of influence itself represents a main focus of interest. Sensitivity analysis on parcel size and density is also performed to assess the competitiveness position of AGVs with lockers.

Costs of ownership (+APK) will increase by 40% over the next eight years due to the termination of current tax discounts for electric vehicles. Yet, it is unclear how strong government support for renewable and green energy investments will be, or whether ownership taxes for internal combustion engine vehicles will be increased. A change in the cost per km as a result of a 20%, 30% and 40% increase in +APK, and a change of 10%, 20% and 30% in the purchasing cost of the vehicle itself will be considered as main components of the TCO per km. A similar sensitivity analysis will be performed for feeding, M&R, insurance and labour costs (see Table 6). Insurance costs for Neolix are the most unpredictable cost segment in terms of direction and amplitude of change. It may either decrease, following the assumption that driverless vehicles will reduce the claim frequency [130], or increase, considering the high costs of sensors needed for autonomy and for repair, at least in early adoption stage. Based on the assumption of a 50% premium reduction, a change of 50%, 100% and 150% in insurance cost will be considered. Changes in cost segments are assessed unidimensionally.

**Table 6.** Sensitivity analysis for cost per km: cost segments change variation.

| Vehicle Name | Ownership (+AKP) | | | Purchasing Cost | | | Feeding | | | Maintenance and Repair | | | Labor Cost | | | Insurance | | |
|---|---|---|---|---|---|---|---|---|---|---|---|---|---|---|---|---|---|---|
| | 20% | 30% | 40% | 10% | 20% | 30% | 10% | 20% | 30% | 10% | 20% | 30% | 10% | 20% | 30% | 50% | 100% | 150% |
| Renault Master Diesel | 0.4% | 0.5% | 0.7% | 1.0% | 2.0% | 3.0% | 4.0% | 5.0% | 6.0% | 3.0% | 4.0% | 5.0% | 10.0% | 16.0% | 23.0% | 2.0% | 4.0% | 7.0% |
| Renault Master Z.E. | 0.04% | 0.06% | 0.09% | 2.0% | 3.0% | 5.0% | 0.4% | 0.7% | 1.0% | 0.2% | 0.5% | 0.7% | 7.0% | 15.0% | 22.0% | 1.0% | 3.0% | 4.0% |
| Neolix Express | 0.006% | 0.01% | 0.2% | 6.0% | 12.0% | 18.0% | 1.0% | 2.0% | 3.0% | 1.0% | 3.0% | 4.0% | 0.0% | 0.0% | 0.0% | 7.0% | 15.0% | 22.0% |

Sensitivity analysis on cost per km confirms the finding of TCO analysis, in which labour cost plays a vital role in total cost formation for non-autonomous vehicles, increasing the cost of kilometre by as much as 23% for Renault Master Diesel and 32% for Renault Master Z.E., in case of a 30% increase in the courier salary.

For autonomous vehicles, the purchasing cost itself represents the cost segment significantly affecting cost per km, giving an 18% increase in total costs with a 30% purchasing cost increase. As for insurance costs, although the percentage change in TCO per km appears large (7% to 22%), it should be noted that it almost coincides with one caused by purchasing cost variation, while the initial change percentages of the purchasing cost segment are 5 times smaller than in the case of the insurance cost segment. Also, in absolute

terms, even 150% of insurance cost variation induced by autonomy is far away from the 10% labour cost variation effect on the TCO per km for the rest of the non-autonomous vehicles, which confirms, from an efficiency perspective, the advantageousness of the next technological transition, even at a potentially high insurance cost.

Since the conditions for using Neolix were equated to cargo bicycles, usually used for express and small parcel delivery, the shift to smaller than 12 L parcels can give more prominent results. Especially since, according to DHL, bicycle couriers can be more productive, delivering up to 25% more parcels than vans in particularly busy and difficult-to-access urban areas [131]. Also, although small-size parcels prevail in B2C deliveries, inefficiency in packaging methods remains a big concern, currently allowing for a 20% reduction in average parcel size according to Jonker and Zschocke (2018) [132]. At the same time, Bol.com plans to ship products with their own sturdy packaging, without outer additional cardboard boxes, encouraging their 3rd party merchants to follow [133]. Retail order consolidation is another practice gaining popularity in e-commerce, which aims to combine multiple orders into one single package, potentially resulting in fewer shipments of fuller cartons [134].

The existing variability in relation to the size and number of packages prompts us to conduct the sensitivity analysis on percentage distributions of 8 L, 13 L, and 18 L parcel dimensions towards the total deliverable capacity of Neolix. Two scenarios are considered: the first puts emphasis on smaller parcels with a percentage distribution of 70%, 20%, and 10%; the second includes Letterbox Parcel + size having a maximum of 3.2 L, instead of big 18 L, with a percentage distribution of 40%, 40%, 20% from the smallest to the biggest parcel size. In the 2nd scenario, the income per parcel delivered is changed according to the Letterbox Parcel + size distributional inclusion and its domestic tariffs (assuming equal distribution among online/offline franking), resulting in a new value of EUR 5.94. Table 7 gives an overview of both average parcel size scenarios, in comparison to the baseline for Neolix, Renault Master Diesel and the electric Z.E. version.

**Table 7.** Sensitivity analysis: average parcel size.

| Renault Master Diesel | Renault Master Z.E. | Neolix | Neolix Scenario 1 | Neolix Scenario 2 | |
|---|---|---|---|---|---|
| 132,880 | 132,880 | 124,254 | 149,742 | 210,276 | Delivery volume per year (units) |
| 78,000 | 78,000 | 103,131 | 103,131 | 103,131 | Delivery kilometer per year (km) |
| 1.70 | 1.70 | 1.20 | 1.45 | 2.04 | Parcels per kilometer delivered (units) |
| 7.00 | 7.00 | 7.00 | 7.00 | 5.94 | Income per parcel delivered (EUR) |
| 11.93 | 11.93 | 8.43 | 10.16 | 12.11 | Income benefit per kilometer (EUR) |
| 0.90 | 0.96 | 0.13 | 0.13 | 0.13 | Cost per kilometer (EUR) |
| 11.03 | 10.96 | 8.30 | 10.03 | 11.98 | Revenue/Loss per kilometer (EUR) |

Scenario 1: The average parcel size in this case is 10 L, resulting in a maximum of 47 parcels that can fit in Neolix (471.5 L). With nine daily delivery cycles, a maximum of 423 daily and 149,742 annual deliveries can be achieved, respectively. Break-even point: 52 parcels of 9.06 L size.

Scenario 2: The average parcel size in this case is 7.08 L, resulting in a maximum of 66 parcels that can fit in Neolix (471.5 L). With nine daily delivery cycles, a maximum of 594 daily and 210,276 annual deliveries can be achieved, respectively. Break-even point: 61 parcels of 7.73 L size.

As expected, with a decrease in the average parcel's size and an increase in Neolix's total capacity, its overall profitability increases. Experimentally, the break-even points in relation to Renault Master Diesel in revenue per km for both scenarios were determined. Another way to improve profitability without changing vehicle capacity is to increase the parcel density of the delivery area, meaning the change in the performance indicator of 0.83 km per parcel delivered, as determined for Neolix. Taking the initial Neolix calculations with 39 parcels capacity and 124,254 annual delivery volume as a baseline, increases of parcel density by 20% in scenario 1 and 30% in scenario 2 were considered. Parcel density directly affects annual millage, meaning that with the higher density, the total km needed to cover this area, and therefore total costs, decrease. Density, on the other hand, can be improved by reducing the overall distance travelled in a single delivery cycle, based on better delivery route planning or deployment of inner-city micro-hubs, reducing the longest plot (depot–first delivery address) of entire delivery cycle steps [96].

Table 8 gives an overview of parcel density scenarios in comparison to Neoli''s baseline, as well as Renault Master in diesel and electric versions. In the first scenario, the increase of 20% gives the new density of 0.69 km per parcel and delivery cycle millage of 26.98 km. The second scenario with a density increase of 30% gives new performance indicators of 0.64 km and 24.90 km, respectively. An improvement of 33% gives a new density of 0.62 and delivery cycle millage of 24.34 km, representing the break-even point from which Neolix brings more revenue per km than both Renault Master vans. As parcel density from the Parcel Market and Home Deliveries analysis was 0.6 km per parcel [96], with additional infrastructure like micro-hubs, Neolix has the potential to be fully competitive.

**Table 8.** Sensitivity analysis: parcel density.

| Renault Master Diesel | Renault Master Z.E. | Neolix | Neolix Scenario 1 | Neolix Scenario 2 | |
|---|---|---|---|---|---|
| 132,880 | 132,880 | 124,254 | 124,254 | 124,254 | Delivery volume per year (units) |
| 78,000 | 78,000 | 103,131 | 85.942 | 79,331 | Delivery kilometre per year (km) |
| 1.70 | 1.70 | 1.20 | 1.45 | 1.57 | Parcels per kilometre delivered (units) |
| 7.00 | 7.00 | 7.00 | 7.00 | 7.00 | Income per parcel delivered (EUR) |
| 11.93 | 11.93 | 8.43 | 10.12 | 10.96 | Income benefit per kilometre (EUR) |
| 0.90 | 0.96 | 0.13 | 0.13 | 0.13 | Cost per kilometre (EUR) |
| 11.03 | 10.96 | 8.30 | 9.99 | 10.83 | Revenue/Loss per kilometre (EUR) |

## 5. Conclusions

The aim of this study was to explore the cost and benefit segments of LMD operations, using AGVs in urban areas. The importance of this work brings novelty to often neglected labour costs of vehicle drivers despite being the biggest cost segment for non-autonomous vehicles, simultaneously representing the potential savings in the case of AGVs. The methodology based on the TCO method based on the TCO analysis was fundamental to the validation of results. Supplementary beneficial factors such as fuel efficiency, total operational time increase, capacity occupation and reliability were evaluated, as opposed to the one-dimensional analysis based solely on costs.

The main results demonstrate the following:

- Purchasing cost was another significant element, proving that technological deployment highly relies on economic feasibility and practicality. Thus, electric vehicles, while still being more expensive than diesel ones, even with governmental subsidies

    (EUR 26,520.95 versus EUR 44,318.75), over an ownership period of 8 years will be cheaper to retain (EUR 25,5894.2 versus EUR 28,0501.96), encouraging an ongoing shift towards electrification. On the contrary, AGVs with lockers, Neolix in this study, despite the lowest purchasing and TCO costs (EUR 24,910.75 and EUR 41,574.82), make autonomy an expensive option, questioning its practicality, as the AG's load capacity is 6 times smaller than a delivery van.

- Regarding fuel efficiency, the effect of the auxiliary tool, SOT, on vehicles' feeding costs was considered, discovering a significant difference in savings capabilities for diesel versus electric vehicles benefiting the former while the cumulative TCO period savings for the latter were not even enough to overcome initial technological investment. Although SOT technology did not bring significant fuel savings for electric vehicles, it still represents interest from a sustainability perspective, being a source of green energy that can otherwise supplement standard charging from a grid or support a cooling system for food delivery.

- Total operational time increase, capacity occupation and reliability benefits affect the total number of parcels delivered, representing the income constituent often neglected in economic assessments. Under the conditions of the maximum capacity of 220 parcels and 39 parcels of 12 L for a delivery van and Neolix, respectively, even with an increase in possible delivery time to 07:00–01:00 as opposed to the traditional 08:00–21:30 range, an AGV is not able to deliver more parcels than a van. Therefore, in sensitivity analysis, factors like average parcel volume and delivery area density were examined, to find out the factors' break-even values when an AGV becomes more profitable than the delivery van. It was discovered that Neolix should be used for parcels with a volume of less than 9.06 L, or in areas of parcel density as high as 0.62 km per parcel, to provide competitive performance results within urban delivery contexts.

- Presumably useful for the practitioners interested in the feasibility of autonomous vehicles for LMD operations, the conducted research proves AGVs are a potentially successful future project considering mandatory urban delivery conditions, high parcel demand density and sufficiently developed infrastructure, that would facilitate the deployment of AGVs similar to cargo bikes. Considering the theoretical implications, this study offers a framework allowing for a monetary-based comparison of vehicles used in LMD, within a multidimensional context of environmental sustainability, effectiveness and efficiency. Furthermore, this paper determines the AGV with lockers as a separate class of innovative solutions along the LMD technological transformation process and makes an academic contribution by investigating its competitiveness as a substitute to the ICEVs and eVs.

    This research has the limitation of the absence of real operational data for AGVs; hence, it utilised, based on reasonable assumptions, data related to the usage of cargo bi-cycles in LMD conditions. A real case study on AGVs would improve the reliability of AGV comparison and is therefore highly advised. Financial benefits for electric vehicles will expire in 2026, which may induce an increase in electric vehicle ownership costs. Therefore, the given research could be repeated in 5 years with modernized vehicle models of all types and their new usage conditions, updated subsidy politics attributed to electric vehicles, and with consideration for possible autonomy-related investment allowances from the Dutch government. It will also allow for a better assessment of the distribution structure of different cost segments and their change patterns over time while simultaneously examining the prerequisites and drivers behind the changes. Further research on SOT technology and its environmental sustainability capabilities is also suggested.

**Author Contributions:** O.L.: writing—original draft, methodology, software, formal analysis, investigation, resources, data curation. A.S.: writing—review and editing, supervision, project administration, funding acquisition, term, conceptualization. All authors have read and agreed to the published version of the manuscript.

**Funding:** The authors would like to thank the OP Zuid program for the funding awarded. The project was made possible in part by financial support from the European Union (European Fund for Regional Development), OP-Zuid, the Province of North Brabant, the Province of Limburg and the Ministry of Economic Affairs and Climate.

**Institutional Review Board Statement:** Not applicable.

**Informed Consent Statement:** Not applicable.

**Data Availability Statement:** Data is contained within the article.

**Conflicts of Interest:** The authors declare no conflict of interest.

## Appendix A. Input Information for a TCO on a Transport Vehicle

TCO assessment formula proposed by Siragusa et al. (2020) [61]:

$$
TCO_T = PP_0 - RV_T + RFC_0 - S_0 \\
+ \sum_{t=0}^{T} \frac{FC_t + IC_t + MRC_t + BC_t + OC_t + RT_t - II_t}{(1+i)^t} \tag{A1}
$$

Ownership period is a predefined time interval parameter that defines the estimation scope (year, life span).

Travel data: Travel statistics or kilometres that a vehicle is predicted to travel during ownership, usually depending on the industry and purpose the vehicle is used for.

Vehicle data cover the cost of the vehicle unit and summarizes its main model characteristics affecting unit price and operational costs. It is the most important and predetermining data set of a TCO, as the vehicle's features and fuel type heavily correlate with vehicle structure and associated costs. For example, to ensure the required level of autonomy, additional equipment is needed, like sensors and cameras to perceive the environment and their own movement, onboard computer hardware and special actuators for vehicle control [135,136], while a driver's presence is usually required in order not to miss some conventional parts. For the same reasons, personnel costs are later included or excluded in operational costs [22]. Fuel type can also entail changes in the form of battery, solar panels, hydrogen tanks or adjusted engine [137], and affect the depreciation rates, changing the residual end value of the vehicle [89];

Cost data traditionally include all operational costs involved, i.e., registration fees, subsidies, feeding (fuel costs), insurance, maintenance and repair, taxes, road toll, indirect incentives that might deviate depending on the economic situation, governmental policies and vehicle-usage-related factors [62].

Registration fee: Any vehicle over 750 Kg must be registered at RDW. The price for this is EUR 10.75 [105].

BPM charge (Bijzondere Verbruiksbelasting van Personenautos) is tax payable by the first person to register a newly purchased vehicle in the Netherlands [138]. However, there is an option to avoid BPM by obtaining the grey license plate that also allows for a reduction benefit on road tax [139]. Both diesel and electric Renault Master delivery vans meet the conditions for the grey license plate when used for business purposes [106], which makes them eligible for exemption from the BPM. Under Dutch law, Neolix is not recognized as a delivery van [140]; therefore, one cannot claim for a grey license plate with Neolix and, from this point in this study, will be equated to a passenger car used for business purposes. For passenger cars, BPM is determined by the $CO_2$ emissions in grams/km; therefore, for electric Neolix, BPM is not charged [107].

APK is a general periodic inspection in Europe. Delivery vans up to 3500 kg obey the same inspection rules as passenger cars [141]. The only difference is the frequency of inspection which depends on the fuel type. Vehicles with electric or gasoline engines need to be first checked when they are four years old, then twice every two years and then every other year (the 4-2-2-1 schedule). For diesel, gas or other engines, the schedule is 3-1-1 [142].

The price of APK varies per service provider and, on average, accounts for EUR 45 for an electric van, EUR 52 for a conventional van and EUR 35 for passenger cars [114,142,143].

Road (motor vehicle) tax is an annual fee for the use of a vehicle that depends on weight, fuel type, level of pollution caused by a vehicle, and province. According to Belastingdienst.nl (2021) [108] conventional diesel vehicles for business use with an "Empty vehicle mass" of 2051–2150 kg should pay EUR 142 every three months, while electric vehicles are exempt from this tax until 2024. Then, in 2025, there will be a 75% tax discount, and from 2026, the full rate of tax should be paid again. The road tax for an electric van with "Empty vehicle mass" of 1951–2050 kg is EUR 135, and for a Neolix type of vehicle (1–550 kg), it is EUR 30, respectively.

VAT is the value-added consumption tax on goods and services. The general VAT in the Netherlands is 21% [144]. However, if a vehicle is fully counted as a business asset, which is the case for this study, the VAT can be deducted [145].

VEHICLE-RELATED COSTS, SUBSIDIES AND INDIRECT INITIATIVES

Subsidy Scheme for Emission-Free Company Cars (SEBA)–subsidy functioning from 15 March 2021 to 31 December 2025 and gives 10% refund up to a maximum EUR 5000 on the net list price for a completely emission-free company vehicle under category N1 [110]. All vehicles under consideration in this research meet the requirement of the N1 category and are therefore eligible for SEBA [146].

Environmental Investment Allowance (MIA) is a tax benefit for investment in environmentally friendly assets approved and mentioned in so-called "positive lists" [111]. Renault Master Z.E., with asset case G3101, is eligible for an allowance for a maximum of EUR 75,000 of the investment amount with a total MIA benefit of 36% and condition of the previous deduction from the purchasing price of any subsidies received like SEBA [147].

Small-scale investment allowance (KIA) involves business investments in assets in the range of EUR 2401 to EUR 59,170 which are are eligible for 28% of the investment amount [109]. There is no evidence that Neolix cannot qualify for this investment allowance; the same can be applicable to Renault Master Z.E. and one with a diesel engine and even to SolarOnTop [148].

Energy investment allowance (EIA) is allowance for $CO_2$ reduction, energy-efficient techniques and sustainable energy-related investments. It gives an average advantage of 11% and lowers energy bills, allowing for the subtraction of the electricity generated with EIA investment [112]. Solar panels used in this research comply with the description under asset code 251115 of Energy List-Solar panels or foil for electricity generation on means of transport (W) [149].

Feeding-Energy or fuel costs, specifically electricity and diesel fuel. Electricity price exc. VAT is 0.11 EUR /kWh [113] and diesel price is EUR 1520/L according to the Average National Recommended Price [115].

Insurance: Auto insurance varies depending on the service provider, and is usually more expensive for businesses than for private vehicles because of higher annual mileage and risks related [150]. Therefore the insurance premiums from Achmea, the leader of the insurance Dutch market [151], were taken. During calculations, the price of Full WA+ Casco was considered based on a 33-year-old driver with no accidents in the last 10 years. Accordingly, for diesel vans, the premium is EUR 128.03 per month, while for electric vans, it is only around EUR 69.84 [152]. Nowadays, nobody can predict the premium amount on autonomous vehicles, especially for delivery robots like Neolix. In this study, an estimation of a 50% premium reduction comparing to the conventional van was chosen, similar to Bösch et al. (2018) and Ongel et al. (2019) [22]. It leads to EUR 64 of the insurance premium for Neolix delivery vehicle.

Maintenance and repair-Maintenance costs for electric vehicles are lower than for a conventional van with an internal combustion engine because they have fewer moving components and do not need oil and filter replacements. Lebeau et al. (2019) [65], in their TCO comparison study on electric vehicles, retrieved the cost of maintenance of 0.019 cEUR /km and 0.052 cEUR /km for electric and diesel vehicles, respectively. Their estimations

are in line with Davis & Figliozzi (2013) [153], claiming that the MR costs of the electric vehicle's battery are half of the conventional cars.

Battery: The battery lifespan is typically determined by a number of full charging cycles performed before the battery reaches a certain level of initial capacity or so-called state of health (SoH). For Renault Master Z.E.'s 33 kW/h battery, the minimum SoH after which it cannot be used in the automotive industry anymore is 66% [154]. After eight years, average battery degradation will reach around 12.8–18.4% [155], still being far from the critical 66% SoH. Battery longevity can vary depending on many factors, for instance, usage, climate, and charging frequency, thus is hard to predict, but according to the Groupe Renault (2021) [156] it was estimated around ten years for its electric vehicle product line. No information was found on the lifespan of the Neolix battery. Therefore, no battery cost will be considered in this research.

Road toll: Currently, there are no toll roads in the Netherlands for vehicles less than 12 tons [157]. However, there are zero-emission zones in 13 municipalities restricting entry of diesel vehicles with emission class 3 and lower [18,158,159] (Milieuzones, 2021). Renault Master with a diesel engine has an emission class of 6.

Annual cost adjustment according to the discount rate of the present discounted value [71]:

$$PV = A_t \times \frac{1}{(1+I)^t} \tag{A2}$$

where

PV = Present value;
$A_t = Amount\ of\ one-time\ cost\ at\ a\ time^t$;
$I = Real\ discount\ rate$;
$T$ = Time (expressed as number of years).

## Appendix B. Vehicle' Descriptive Information

**Technical details**

| | |
|---|---|
| Model: | SLV11 |
| Drive system: | electric (battery) |
| Range per battery charge: | 100 kilometres |
| Speed in the trial: | 6 km/h |
| Range in hours: | around 8 hours |
| Unladen weight: | 371 kg |
| Maximum load: | 350 kg |
| Length x width x height: | 2.69 x 1 x 1.87 metres |
| | |
| Number of compartments: | 22 (2×11) |
| Compartment dimensions: | large compartments: 510 x 280 x 420 mm / small compartments: 125 x 245 x 420 mm |
| | |
| Turning radius: | 6.90 metres |
| Braking distance: | around 1 metre at 10 km/h (depends on speed) |

**Battery**

| | |
|---|---|
| Charging time: zero hours | (battery can be replaced) |
| Capacity: | 12.9 kw/h |
| Voltage: | 72 volts |
| Maximum power: | 8 kilowatts |

**Safety/navigation systems**
4 LiDAR systems, 6 cameras and 14 sensors

**Figure A1.** Neolix specifications and details. Dérobert (2020) [76].

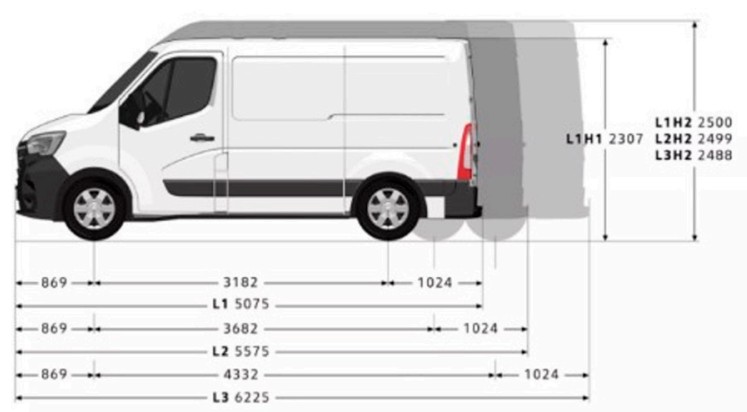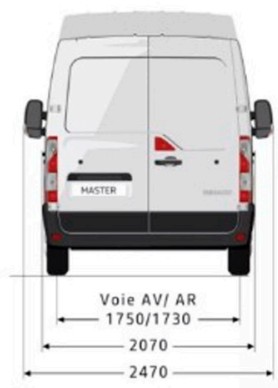

**Figure A2.** Renault Master specifications and details. Sources: Business.renault.co.uk (2021) [78].

## Appendix C. Purchasing and Annual Ownership (+APK) Costs Calculations

**Table A1.** Purchasing cost calculations.

| Price Segments/Vehicle | Master Diesel | Master Diesel + SolarOnTop | Master Z.E. | Master Z.E. + SolarOnTop | Neolix | Neolix + SolarOnTop |
|---|---|---|---|---|---|---|
| Price of vehicle excl. VAT/BPM | 31,940.00 | 31,940.00 | 58,700.00 | 58,700.00 | 30,000.00 | 30,000.00 |
| Price of SolarOnTop | | 5350.00 | | 5350.00 | | 5250.00 |
| **Vehicle related capital costs** | | | | | | |
| Registration fee | 10.75 | 10.75 | 10.75 | 10.75 | 10.75 | 10.75 |
| BPM 37.7% (+273) | | | | | | |
| Total | 31,950.75 | 37,300.75 | 58,710.75 | 64,060.75 | 30,010.75 | 32,260.75 |
| **Subsidies and Indirect initiatives** | | | | | | |
| MIA 36% (9%) | — | — | 21,132.00 | 21,132.00 | | |
| KIA 28% (5.5%) | 8943.20 | 10,441.20 | 16,436.00 | 17,934.00 | 8400.00 | 9870.00 |
| EIA 45.5.% (11%) | — | 2434.25 | — | — | — | 2388.75 |
| Net benefit at 25% nominal tax | 2235.80 | 3218.86 | 9392.00 | 9766.50 | 2100.00 | 3064.69 |
| SEBA | 3194.00 | 3194.00 | 5000.00 | 5000.00 | 3000.00 | 3000.00 |
| Total subsidies | 5429.80 | 6412.86 | 14,392.00 | 14,766.50 | 5100.00 | 6064.69 |
| **Total purchasing price** | **26,520.95** | **30,887.89** | **44,318.75** | **49,294.25** | **24,910.75** | **29,196.06** |

Sources—RDW [105], Autodijk.nl [114], Belastingdienst [109,138,144,145,148], RVO [110,112,147,149], Renault.nl [78], Japan-times.co.jp [81], Imefficiency [84].

**Table A2.** Annual Ownership (+APK) costs calculations.

| Year | | Master Diesel | Master Diesel + SolarOnTop | Master Z.E. | Master Z.E. + SolarOnTop | Neolix | Neolix + SolarOnTop |
|---|---|---|---|---|---|---|---|
| 2021 | 1 | 567.72 | 567.72 | | | | |
| 2022 | 2 | 567.43 | 567.43 | | | | |
| 2023 | 3 | 619.07 | 619.07 | | | | |
| 2024 | 4 | 618.76 | 618.76 | 44.91 | 44.91 | 34.93 | 34.93 |
| 2025 | 5 | 618.45 | 618.45 | 33.67 | 33.67 | 7.48 | 7.48 |
| 2026 | 6 | 618.14 | 618.14 | 179.46 | 179.46 | 64.81 | 64.81 |
| 2027 | 7 | 617.83 | 617.83 | 134.53 | 134.53 | 29.90 | 29.90 |
| 2028 | 8 | 617.53 | 617.53 | 179.28 | 179.28 | 64.74 | 64.74 |
| **Total 8 years** | | **4844.94** | **4844.94** | **571.85** | **571.85** | **201.85** | **201.85** |

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
