# Peer review of "Efficiency in the Last Mile of Autonomous Ground Vehicles with Lockers: From Conventional to Renewable Energy Transport"

_sustainability, doi:10.3390/su152316219_

Round 1

Reviewer 1 Report

Comments and Suggestions for Authors

1. A "Nomenclature" is needed.
2. The quality of figures 2 to 5 is low.
3. All price data within the tables must be accurately referenced.
4. There are many typographical errors. For example, the number 2 in CO2 must be lowercase.
5. Why was no 2023 reference used in this regard?
6. The conclusion section is very long and lacks proper structure. The importance and necessity of doing work, methodology, investigated parameters, innovation of work, expression of results in bullet-point form, etc. constitute a suitable structure.

Comments on the Quality of English Language

Minor editing of English language required

Author Response

Reviewer #1:

Reviewer #1:

  1. A "Nomenclature" is needed.

Revision: Thank you. Sorry, we did not understand this query. We will be pleased to make this review after some more clarification: Which kind of nomenclature and where should it be added?

  1. The quality of figures 2 to 5 is low.

Revision: We have replaced the figure with better quality ones.

  1. All price data within the tables must be accurately referenced.

Revision: We have now added source of price data.

  1. There are many typographical errors. For example, the number 2 in CO2 must be lowercase.

Revision: We have made this correction as following: CO2

  1. Why was no 2023 reference used in this regard?

Revision: The research was made prior to 2023 thus there was not publications yet. We have updated with some publications adding these references:

 Ensafian H, Zare Andaryan A, Bell MGH, Glenn Geers D, Kilby P, Li J. Cost-optimal deployment of autonomous mobile lockers co-operating with couriers for Simultaneous Pickup and Delivery Operations. Transp. Res. C: Emerg. Technol. 2023, 146, 1–2, doi:10.1016/j.trc.2022.103958.

Fehling, C. Saraceni, A. Technical and legal critical success factors: Feasibility of drones & AGV in the last-mile-delivery, Research in Transportation Business & Management. 2023, 50, 101029, https://doi.org/10.1016/j.rtbm.2023.101029

Kumar, D., Kalghatgi, G., Agarwal, A.K. Comparison of Economic Viability of Electric and Internal Combustion Engine Vehicles Based on Total Cost of Ownership Analysis. In: Upadhyay, R.K., Sharma, S.K., Kumar, V., Valera, H. (eds). ITMS. 2023, 455–489, https://doi.org/10.1007/978-981-99-1517-0_20

Angamarca-Avendaño, D., Saquicela-Moncayo, J., Capa-Carrillo, B., & Cobos-Torres, J. (2023). Charge Equalization System for an Electric Vehicle with a Solar Panel. Energies, 2023, 16(8), 3360, https://doi.org/10.3390/en16083360

  1. The conclusion section is very long and lacks proper structure. The importance and necessity of doing work, methodology, investigated parameters, innovation of work, expression of results in bullet-point form, etc. constitute a suitable structure.

Revision: Thank you for this suggestion. We have now re-structured the Conclusion section in bullet-point form with proper highlights about the importance of the work methodology, investigated parameters, innovation of work.

Reviewer 2 Report

Comments and Suggestions for Authors

The article presents a very good approach of Energy efficiency in the Last Mile: From conventional to renewable energy transport. The article is well written. But it can be improved in terms of presentation to attract the readers:

1- Section 2 needs a table of comparision in aspects of optimization methods, Efficency, and cost.

2- Fig 2 need to be represented in better quality. 

3-  Section 3 is out of any equation!. Even though, susection 3.2 discusses the benifits of calculations. Therefore, some of equations need to be presented in this section rather than Appendix.

4- The conlusion is quite long. It should be focuused on the main findings. Or atleast in points.

Comments on the Quality of English Language

The article is well written. May authors double check efore revision to avoid any language issues.

Author Response

Reviewer #2:

Reviewer #2: The article presents a very good approach of Energy efficiency in the Last Mile: From conventional to renewable energy transport. The article is well written. But it can be improved in terms of presentation to attract the readers:

  • Section 2 needs a table of comparision in aspects of optimization methods, Efficency, and cost.

Revision: Thanks for point it out. Table 1 includes the overview of the information that will later be needed to define various main segments of cost-benefit analysis. Purchasing price and curb weight are used to assess eligibility for subsidies as well as define the exact rate of various ownership costs introduced further in the paper. Motor type data together with vehicle fuel consumption is used in calculations to determine the fuel efficiency impact (cost savings) provided by SolarOnTop (SOT) technology. Payload, maximum speed, and maximum range serve as input to calculate benefit or maximum daily delivery volumes per vehicle. We have now improved this clarification in the paper.

  • Fig 2 need to be represented in better quality.

Revision: We have replaced the figure 2 with better quality.

  • Section 3 is out of any equation!. Even though, susection 3.2 discusses the benifits of calculations. Therefore, some of equations need to be presented in this section rather than Appendix.

Revision: The equations were moved to Section 3.

  • The conlusion is quite long. It should be focuused on the main findings. Or atleast in points.

Revision: Thank you for this suggestion. We have now re-structured the Conclusion section in bullet-point form with proper highlights about the importance of the work methodology, investigated parameters, innovation of work.

Comments on the Quality of English Language

The article is well written. May authors double check efore revision to avoid any language issues.

Revision: Quality of English Language was revised.

Reviewer 3 Report

Comments and Suggestions for Authors

This paper mainly discusses an idea about “autonomous ground vehicles with lockers”. Overall, the topic is interesting and important. The authors are advised to consider the following suggestions to further improve the paper quality.

(1) The title needs to be changed at least to show the idea of the paper, such as including “autonomous ground vehicles with lockers”

(2) The introduction section require further expanded demonstrate the attractiveness of the idea of “autonomous ground vehicles with lockers”

(3) Some the statements in the paper require further support from existing studies. The authors are suggested to add SAE Technical Paper 2023-01-0192, SAE Technical Paper 2023-01-0334, Journal of Energy Resources Technology, 2023; 145(1): 012302, Journal of Energy Resources Technology, 2022; 144(11): 112307, ASME ICEF2022-88682 to the reference list.

(4) The figure quality of all figures needs to be further improved. Some of the words in the figure is too small.

(5) The appendix seems to have business information. Usually the paper should avoid advertisement.

(6) The writing needs to be further polished. Some sentences still have grammar mistakes and typo errors.

Comments on the Quality of English Language

The writing needs to bfuther polished.

Author Response

Reviewer #3:

Reviewer #3: This paper mainly discusses an idea about “autonomous ground vehicles with lockers”. Overall, the topic is interesting and important. The authors are advised to consider the following suggestions to further improve the paper quality.

  • The title needs to be changed at least to show the idea of the paper, such as including “autonomous ground vehicles with lockers”

Revision: We have updated the title to “Efficiency in the Last Mile of autonomous ground vehicles with lockers: From conventional to renewable energy transport”.

  • The introduction section require further expanded demonstrate the attractiveness of the idea of “autonomous ground vehicles with lockers”

Revision: Arguments on attractiveness of “autonomous ground vehicles with lockers” were added to the introduction section. 

  • Some the statements in the paper require further support from existing studies. The authors are suggested to add SAE Technical Paper 2023-01-0192, SAE Technical Paper 2023-01-0334, Journal of Energy Resources Technology, 2023; 145(1): 012302, Journal of Energy Resources Technology, 2022; 144(11): 112307, ASME ICEF2022-88682 to the reference list.

Revision: We have added some publications to support the statement, adding these references:

Srivastava, S.; Karthikeyan, S.; Arumugam, P.; Kumar, A.; Thanigaivel, G. Design, Development and Experimental Investigation on the Effect of HVAC Power Consumption in Electric Vehicle Integrated with Thin Film Solar PV Panels. SAE Tech. Pap, 2021, https://doi.org/10.4271/2021-28-0122.

Angamarca-Avendaño, D., Saquicela-Moncayo, J., Capa-Carrillo, B., & Cobos-Torres, J. (2023). Charge Equalization System for an Electric Vehicle with a Solar Panel. Energies, 2023, 16(8), 3360, https://doi.org/10.3390/en16083360

Ensafian H, Zare Andaryan A, Bell MGH, Glenn Geers D, Kilby P, Li J. Cost-optimal deployment of autonomous mobile lockers co-operating with couriers for Simultaneous Pickup and Delivery Operations. Transp. Res. C: Emerg. Technol. 2023, 146, 1–2, doi:10.1016/j.trc.2022.103958.

Parcel delivery. The future of last mileAvailable online https://www.bringg.com/wp-content/uploads/2016/10/Parcel_delivery_The_future_of_last_mile-1.pdf (accessed on 23 September 2023).

Fehling, C. Saraceni, A. Technical and legal critical success factors: Feasibility of drones & AGV in the last-mile-delivery, Research in Transportation Business & Management. 2023, 50, 101029, https://doi.org/10.1016/j.rtbm.2023.101029

Saraceni, A., Oleko, R., Guan, L., Bagaria, A., Quintens, L. Autonomization and Digitalization: Index of Last Mile 4.0 Inclusive Transition. In: Kim, D.Y., von Cieminski, G., Romero, D. (eds) Advances in Production Management Systems. Smart Manufacturing and Logistics Systems: Turning Ideas into Action. APMS 2022. IFIP Advances in Information and Communication Technology, 2022, v. 663. Springer, Cham. https://doi.org/10.1007/978-3-031-16407-1_21

  • The figure quality of all figures needs to be further improved. Some of the words in the figure is too small.

Revision: We have replaced the figures with better quality ones.

  • The appendix seems to have business information. Usually the paper should avoid advertisement.

Revision: We have added the data sources for our research. We now have tried to remove any part that was elaborated as advertisement.

  • The writing needs to be further polished. Some sentences still have grammar mistakes and typo errors.

Revision: Quality of English Language was revised.

Comments on the Quality of English Language

The writing needs to bfuther polished.

Revision:  Quality of English Language was revised.

Reviewer 4 Report

Comments and Suggestions for Authors

Review

This paper compares conventional and electric vans, from the point of view of associated vehicle costs, benefits related to the use, and economic feasibility taking into consideration the cost per kilometer.

The paper is interesting.

However, the aim of this paper and the research part are not clear. They must be clearly stated in the Abstract and in the Introduction.

The technical data of the three kinds of vehicles (from Table 1) are not complete. To obtain a convincing comparison between the vehicles, for each vehicle must be added the full set of technical indexes regarding its technical performance such as power, torque, rated speed, maximum speed, acceleration, efficiency, etc. Then must clearly show which characteristics are similar. It is not correct to compare totally unequal things.

The Figures 1-5 and the associated letters and numbers are not visible and cannot be read. The size and quality of the letters must be improved.

The section Conclusions is too big and must be restructured: fragment and separate the paragraphs, clarify the statements, move most text to a separate section Discussion, then keep in Conclusions the most important facts.  

Comments on the Quality of English Language

English language must be edited for syntax, style and vocabulary.

Author Response

Reviewer #4:

Reviewer #3: This paper compares conventional and electric vans, from the point of view of associated vehicle costs, benefits related to the use, and economic feasibility taking into consideration the cost per kilometer. The paper is interesting. However, the aim of this paper and the research part are not clear. They must be clearly stated in the Abstract and in the Introduction.

Revision: Thank you for point this out. We have improved the clarity of the research aim at both Abstract and Introduction section.

The technical data of the three kinds of vehicles (from Table 1) are not complete. To obtain a convincing comparison between the vehicles, for each vehicle must be added the full set of technical indexes regarding its technical performance such as power, torque, rated speed, maximum speed, acceleration, efficiency, etc. Then must clearly show which characteristics are similar. It is not correct to compare totally unequal things.

Revision: Table 1 includes the overview of the information that will later be needed to define various main segments of cost-benefit analysis. Purchasing price and curb weight are used to assess eligibility for subsidies as well as define the exact rate of various ownership costs introduced further in the paper. Motor type data together with vehicle fuel consumption is used in calculations to determine the fuel efficiency impact (cost savings) provided by SolarOnTop (SOT) technology. Payload, maximum speed, and maximum range serve as input to calculate benefit or maximum daily delivery volumes per vehicle. We have now improved this clarification in the paper.

The Figures 1-5 and the associated letters and numbers are not visible and cannot be read. The size and quality of the letters must be improved.

Revision: We have replaced the figures with better quality ones.

The section Conclusions is too big and must be restructured: fragment and separate the paragraphs, clarify the statements, move most text to a separate section Discussion, then keep in Conclusions the most important facts.

Revision:  Thank you for this suggestion. We have now re-structured the Conclusion section in bullet-point form with proper highlights about the importance of the work methodology, investigated parameters, innovation of work.

Comments on the Quality of English Language

English language must be edited for syntax, style and vocabulary.

Revision: Quality of English Language was revised.

Round 2

Reviewer 4 Report

Comments and Suggestions for Authors

The revised manuscript can advance for publication.